# Ambitions and realities: Are Global Fund investments designed to achieve resilient and sustainable systems for health? Findings from the Global Fund Prospective Country Evaluation

**Nicole Salisbury**[1]*, **Saira Nawaz**[1], **Justine Abenaitwe**[2], **Audrey Batzel**[3], **Emily Grapa**[1], **Francisco Rios Casas**[4], **Virginia Cerezo**[3], **Matthew Cooper**[5], **Herbert C. Duber**[4], **Ibrahima Gaye**[6], **Bernardo Hernandez**[4], **Constant Kingongo**[7], **Louisiana Lush**[5], **Kate Macintyre**[5], **Eugene Manika**[7], **Shakilah N. Nagasha**[2], **Tidiane Ndoye**[6], **Rosario Orozco**[3], **Allison Osterman**[1], **Katharine D. Shelley**[1]

1 Health Systems Team, Primary Health Care Program, PATH, Seattle, Washington, United States of America, 2 Infectious Diseases Research Collaboration (IDRC), Kampala, Uganda, 3 Centro de Investigación Epidemiológica en Salud Sexual y Reproductiva (CIESAR), Guatemala City, Guatemala, 4 Department of Health Metrics Sciences, Institute for Health Metrics and Evaluation, Seattle, Washington, United States of America, 5 Euro Health Group, Copenhagen, Denmark, 6 Institut de Santé et Développement (ISED), Université Cheikh Anta Diop (UCAD), Dakar, Senegal, 7 DRC Country Program, PATH, Kinshasa, Democratic Republic of the Congo

* nsalisbury@path.org

**Data Availability Statement:** Quantitative analyses were based on third-party data (detailed grant

## Abstract

Strengthening resilient and sustainable systems for health (RSSH) is central to the Global Fund's strategy, however questions persist about the Global Fund's role in the health systems strengthening space, and the extent to which investments are designed to achieve strengthening objectives, or just fill in gaps in the system. This paper reports on findings from the Prospective Country Evaluations (PCE), a multi-country multi-year evaluation of Global Fund support. We adapted a framework from Chee et al. (2013) to assess whether Global Fund investments were designed to strengthen or support the health system. Per this framework, 'systems support' refers to improvements in health systems functioning primarily driven by increases in inputs, whereas 'systems strengthening' refers to activities that drive changes in how the health system operates (often related to policies, regulations, governance structures, behavior change, and resource optimization). In the 2017 and 2019 funding cycles, we found that despite calls from the Global Fund to invest more strategically to strengthen health systems, a high proportion of RSSH funding was directed toward activities that support the health system. Factors underlying this pattern include limitations imposed by the three-year grant cycle, a lack of clear guidance on how to design strengthening investments, a persistent need for funding to address input gaps, and minimal feedback during the funding request process related to RSSH design. For the Global Fund, and indeed other global health initiatives, to contribute to sustained strengthening of health

budgets) accessed directly from the Global Fund and the authors are not legally able to distribute this data set. While detailed budgets are not presently publicly available on the Global Fund's website, data requests may be directed to the Global Fund's Independent Evaluation Panel (which has replaced the Technical Evaluation Reference Group) at +4158 791 1700 or at https://www.theglobalfund.org/en/contact/. The qualitative data transcripts that support the findings cannot be shared publicly per the terms under which respondents were consented to participate. Additionally, per the terms of our IRB approvals and informed consent process, we did not share primary qualitative data with the Global Fund, or any other external entity. Because the Global Fund personnel overseeing the evaluation are no longer at the Global Fund, it is not possible at this stage to grant them access to the analysis tables so that they can field any such requests. As such, we are unable to provide non-author contacts for access to analysis tables. However, analysis tables may be available upon reasonable request to the authors and with the permission of all organizations within the research consortia and permission of the Global Fund.

**Funding:** The Global Fund PCE was commissioned in 2017 by the Global Fund's Technical Evaluation Reference Group (TERG), an independent evaluation advisory group that assessed and reported on the monitoring and evaluation work conducted by the Global Fund Secretariat, and all authors included in this manuscript were supported financially by the Global Fund PCE grant (TGF-16-144). Members of the Global Fund TERG provided input on the study design, framing of the evaluation questions, and reviewed this manuscript, however, the views in this paper do not necessarily represent the views of the TERG or of the Global Fund Secretariat.

**Competing interests:** The authors have declared that no competing interests exist.

systems, is likely to require enhanced guidance and technical assistance, as well as improved measurement of outputs and outcomes.

## Introduction

In the last two decades, global health initiatives (GHIs) have emerged as a central mechanism through which financial resources are raised and disbursed to low and middle-income countries to support public health programming. Many GHIs were launched with a primary focus on specific diseases or health areas through vertical funding streams. However, in more recent years GHIs have placed increased emphasis on health systems strengthening (HSS), largely out of the recognition that robust health systems are required to achieve ambitious and long-term disease-specific objectives, improving morbidity and mortality across diseases and improving the efficiency of investments.

Much debate persists about whether and how GHIs sufficiently prioritize HSS in relation to their typically much larger portfolio of core health area-specific funding [1–3], as well as their propensity to skew national priorities and resources in favor of focus diseases, sometimes at the expense of broader, cross-cutting efforts to strengthen systems for health. [4]. These debates reflect a wide range of interpretations surrounding what constitutes health systems strengthening. Acknowledging that much donor funding for HSS is actually directed toward gap-filling and addressing input constraints, Chee et al. called for the need to distinguish more explicitly between activities that *support* the health system from activities that *strengthen* the health system [5]. According to Chee et al.

> 'Supporting the health system can include any activity that improves services, from upgrading facilities and equipment to distributing mosquito nets to promoting health behaviors. These activities improve the system's functionality primarily by increasing inputs and can be short term and narrowly focused… In contrast, strengthening the health system is accomplished by more comprehensive changes to policies and regulations, organizational structures, and relationships across the health system building blocks that motivate changes in behavior, and/or allow more effective use of resources to improve multiple health services [5 p86].'

While both are necessary, Chee and colleagues emphasize that failing to invest in strengthening activities may undermine calls for more investment in HSS, because if designed to support and not strengthen, HSS investments are unlikely to achieve their objectives. In more recent years, and particularly in light of the COVID-19 pandemic and the increased emphasis on health systems resilience, there is growing acknowledgement that health systems support too is essential in addressing acute gaps and laying the groundwork for longer-term efforts aimed at strengthening resilient health systems. Indeed, the WHO Toolkit on Health Systems Resilience notes that 'operationalizing health systems resilience involves a wide array of system elements that are required to connect and work in unison, to shift from the concept to the application of resilience in all countries. Therefore, all efforts to make health systems resilient must proactively apply an integrated approach and focused actions using systems thinking [6].' A recent position paper by UHC2030 reinforces this point, noting that taking a systems perspective to health systems strengthening can 'support achievement of multiple health

outcomes and bring together distinct agenda, such as universal health coverage and health security and/or a focus on multiple disease priorities [7].'

Established in 2002, the Global Fund to Fight AIDS, Tuberculosis and Malaria (hereafter, the Global Fund) is one of the largest GHIs, providing over $4 billion a year in funding to halt the spread of HIV, tuberculosis and malaria [8]. Shortly after its founding, the Global Fund added to its portfolio funding to reinforce weak health systems, in recognition that investments to achieve disease specific objectives were too often hampered by weak health systems. The Global Fund's approach to financing HSS has evolved over the years, from a specific call for HSS grants during its fifth funding round to the current funding model (also known as the 'new funding model') which allows for both stand-alone grants for cross-cutting investments in RSSH [2], and/or for RSSH investments to be embedded within larger disease-specific grants. Today, the Global Fund represents a significant HSS donor, and even identifies itself as the world's largest multilateral provider of funding for HSS [9]. The prominence of HSS in relation to the Global Fund's organizational mission has increased over the last decade, and in its 2017–2022 strategy [10], building resilient and sustainable systems for health (RSSH) was one of four strategic objectives. According to the Global Fund,

> 'Resilient and sustainable system for health are the foundation of healthy, productive communities. RSSH is not just about government health systems, but also about services provided by communities, the private sector and other providers, which together should ensure that peoples' health needs are met wherever they seek care. These systems are essential for ending HIV, TB and malaria as public health threats, producing better health outcomes for all and delivering health services in a sustainable, equitable and effective way [11].'

In the most recent 2023–2028 strategy, the Global Fund seeks to 'maximize people-centered integrated systems for health to deliver impact, resilience and sustainability' by supporting the delivery of integrated people-centered quality services, strengthening community systems, data systems and supply chains, and engaging the private sector, among other things [12]. As a result, its overall contribution to RSSH has increased steadily between funding cycles, from $3.27 billion in the 2017–2019 funding cycle to $4.9 billion in the 2020–2022 funding cycle and is projected to rise to $6 billion in the 2024–2026 funding cycle [13].

However, studies have noted that much of the Global Fund's investment in HSS is dedicated to disease specific rather than system-wide interventions [2]. As such, in recent years, the Global Fund Secretariat has been examining how and where the Global Fund invests its RSSH resources, particularly in the lead up to the sixth replenishment which called for doing things differently to change trajectory to end the epidemics of the three diseases and contribute to the sustainable development goals. A 2018 report by the Global Fund's Technical Review Panel (TRP) adapted Chee's 2013 framework and applied it in an analysis of RSSH investments from 16 funding requests submitted in the 2017–2019 funding cycle [14]. Expanding the framework to 4S (systems start-up, systems support, systems strengthening and systems sustainability) the TRP found that most funding for RSSH in the 2017–2019 funding cycle was directed toward systems support, and called for greater investment in systems strengthening and sustainability [14, 15]. In this vein, the Global Fund released new RSSH guidance notes in 2019 and 2020 to support countries in making 'better' RSSH investments in the 2020–2022 funding cycle and implemented regional RSSH workshops leading up to the funding request development process [16, 17]. The RSSH Information Note includes 8 pre-determined modules in which countries can invest, including: Health products management and systems strengthening; Health management information systems and M&E; Human resources for health including community health workers; Integrated service delivery and quality

improvement; Financial management systems; Health sector governance and planning; Community systems strengthening; and Laboratory systems [16, 17]. Furthermore, the Information Note specifies "While the Global Fund recognizes that a mix of RSSH investments is needed given the diverse needs at the country level, applicants are encouraged to shift from a focus on short-term, input focused support (such as vehicles, travel, training costs, equipment, and others) towards more strategic investments (such as strengthening management, improving accountability mechanisms, empowering service providers, and others) that build capacity and lead to sustainable results."

The Global Fund Prospective Country Evaluation (hereafter the PCE), a multi-country, mixed methods evaluation implemented from 2017–2021, aimed to assess how the Global Fund business model factors influence grant design, implementation, and performance. Building on the analysis of the TRP, the PCE applied Chee's framework to compare final grant awards in the 2017–2019 funding cycle (also referred to as new funding model 2, or NFM2) with the 2020–2022 funding cycle (also referred to as new funding model 3 or NFM3) to determine whether a shift toward more strategic investment in activities to strengthen the health system had indeed taken place. Acknowledging the need to understand the complex mix of factors (including the frequently cited need to address acute gaps in the system) that drive decision making around RSSH investments, we augmented this quantitative exercise with a qualitative exploration of the investments themselves.

## Materials and methods

In 2017 The Global Fund's Technical Evaluation Reference Group (TERG) commissioned the PCE (see Box 1 for a description of the full suite of methods employed by the PCE) to prospectively evaluate the implementation, effectiveness and impact of the Global Fund's 2017–2022 strategy [18].

This manuscript describes one analysis embedded within the larger PCE platform and relied upon a subset of the methods described in Box 1. Specifically, this analysis drew upon a mix of quantitative and qualitative methods, including analysis of Global Fund budget data to quantify the level of investment allocated to different supportive and strengthening RSSH activities, document review of Global Fund funding request narratives and key informant interviews to help contextualize the quantitative data and describe why funds were allocated the way they were.

### Design

The analysis of RSSH investments was conducted in seven PCE countries, representing a variety of Global Fund portfolio types (Table 1).

A working group of global evaluation partners (GEPs) and country evaluation partners (CEPs) co-designed an evaluation protocol outlining our approach (S1 Text). Based on the work of Chee and the Global Fund TRP, three criteria were adopted to determine whether a RSSH intervention supports or strengthens the health system: scope, longevity and approach (Table 2) [5, 14].

### Data collection

Table 3 outlines the data collection methods employed in this analysis. Quantitative RSSH data was provided by the Global Fund Secretariat and was extracted from Global Fund final grant award budgets from each country for NFM2 and NFM3 into an analysis file in Excel. The timing of Guatemala Global Fund grants did not align with that of other PCE countries. At the time of analysis, Guatemala had only submitted a NFM3 funding request, and received

## Box 1. Global Fund PCE methods

The Global Fund Prospective Country Evaluation was an independent, prospective multi-year evaluation of the Global Fund in eight countries. The PCE was comprised of two evaluation consortiums, one led by the Institute for Health Metrics and Evaluation and PATH, and the second led by Euro Health Group, in collaboration with evaluation partners in the DRC, Guatemala, Senegal, Uganda, Cambodia, Mozambique, Myanmar and Sudan. The objective of the PCE was to generate actionable insights to accelerate progress toward achieving the Global Fund's strategic objectives through evaluation of the application of the Global Fund business model in the eight PCE countries.

The PCE employed a mixed methods approach, triangulating across a variety of qualitative and quantitative data sources and analytic approaches. Primary data collection methods included meeting observations, in-depth key informant interviews and brief fact-checking interviews. Secondary data sources included Global Fund funding requests, detailed budgets for active and planned grants, implementation letters and progress update/disbursement requests, among others. Specific evaluation protocols and tools were tailored by country evaluation partners to each country's context. Analytic approaches included budget variance analysis, root cause analyses, indicator performance tracking and cross-country synthesis. The analysis presented in this paper relied upon a subset of these methods, described below.

approval, for their HIV grant. As such, for Guatemala we only compare the NFM2 and NFM3 final grant award budgets for HIV, whereas for other countries we included final approved grant budgets for HIV, TB and malaria. Qualitative data was collected through document review and key informant interviews. Relevant documents were provided by the Global Fund TERG and included Global Fund funding request narratives for the 2017–2019 (NFM2) and the 2020–2022 (NFM3) funding cycles, and feedback from the Global Fund TRP on funding requests. Document review provided an overview of the design of RSSH investments in PCE

**Table 1. Overview of Global Fund portfolios in PCE countries.**

| Global Fund Portfolio Characteristics | Cambodia | DRC | Guatemala | Mozambique | Myanmar | Senegal | Sudan | Uganda |
|---|---|---|---|---|---|---|---|---|
| Portfolio Type in 2017–2019 allocation period[#] | High Impact | High Impact | Core | High Impact | High Impact | Core | High Impact | High Impact |
| World Bank income category (2017) † | Low Income | Low Income | Upper LMI | Low income | Lower LMI | Lower LMI | Lower LMI | Low Income |
| HIV burden † | High | High | High | Extreme | High | High | Low | Severe |
| Tuberculosis burden † | Severe | Severe | Moderate | Severe | Severe | High | Moderate | Severe |
| Malaria burden † | Severe | Extreme | Moderate | Extreme | Severe | High | High | Extreme |
| Allocation 2017–19 (US$, millions)[13], total | 83.5 | 527.1 | 32.0 | 502.9 | 262.2 | 71.5 | 129.6 | 465.1 |
| RSSH across signed grants^ (US$, millions), total | 2.3 | 68.9 | 4.4 | 32.8 | 26.8 | 12.4 | 11.3 | 5.5% |
| RSSH across signed grants^, proportion | 4.2% | 12.0% | 13.9% | 6.3% | 11.9% | 16.4% | 8.5% | 1.1% |

[#]Portfolio categories are primarily defined by overall size of the allocation and risk profile.

†Both income level and disease burden determine country eligibility for Global Fund funding and portfolio categorization. Income level eligibility is based on the World Bank Income Classifications; disease burden data are provided by WHO and UNAIDS [19].

^Total RSSH funding is summarized across a country's grants, and includes "direct" RSSH only, meaning investments that were categorized as RSSH per the Global Fund's modular framework definitions [20].

**Table 2. Criteria of supportive or strengthening interventions.**

| Parameter | System Support | System Strengthening |
|---|---|---|
| **Scope** | May be focused on a single disease or intervention | Activities have impact across health services and outcomes; and systems may be integrated into the overall health system |
| **Longevity** | Effects limited to period of funding | Effects will continue after funded activities end |
| **Approach** | Provide inputs to address identified system gaps | Revise policies and institutional relationships to change behaviors and resource use to address identified constraints in a more sustainable manner |
| **Example** | Purchases of vehicles, fuel, and maintenance costs | Developing protocols for data quality monitoring and data review meetings |

countries, how those investments evolved over the successive grants and contextualized them within the larger Global Fund investment in disease programs, informing the designation of support and strengthening for specific activities in the RSSH budget files. Quantitative data and qualitative documents were accessed for this analysis between July 1, 2019 and October 31, 2020.

We conducted two phases of key informant interviews between July 1, 2019 and October 31, 2020, across all evaluation sites, with a range of country and global-level stakeholders. The first phase of key informant interviews (N = 22 total) was conducted in Guatemala (N = 8), Senegal (N = 7) and Uganda (N = 7) and focused on RSSH modules with a high level of funding and/or notable change in investment patterns between funding cycles. Respondents included representatives of Global Fund principal recipients, ministries of health and Global Fund country teams. Drawing on emerging findings from the quantitative analysis of budget data, CEPs developed interview guides probing on change in RSSH investment patterns from NFM2 to NFM3.

The second phase of interviews (N = 76 total) was conducted with a range of in-country and global-level RSSH stakeholders (Cambodia N = 8; DRC N = 12; Guatemala N = 7; Mozambique N = 12; Myanmar N = 18; Senegal N = 7; Uganda N = 10; Global N = 2) to understand how and why decisions were made surrounding the design of RSSH investments. Respondents included ministry of health officials, representatives of Global Fund principal recipients,

**Table 3. Data collection methods.**

| Objective | Method | Data Source |
|---|---|---|
| Determine whether there was evidence of a shift in how the Global Fund invests its RSSH resources | Document Review | Global Fund Funding Request Narratives NFM2 and NFM3 |
| | Budget Analysis | Global Fund final grant award budgets NFM2 and NFM3 |
| Explore the factors underlying the shift (or lack thereof) | Phase 1 Key informant interviews to examine the content and implementation of RSSH modules with high levels of investment in a subset of PCE countries | Guatemala N = 8 Senegal N = 7 Uganda N = 7 |
| | Phase 2 Key informant interviews to understand the factors influencing decision-making about how to design RSSH investments | Cambodia N = 8 DRC N = 12 Guatemala N = 7 Mozambique N = 12 Myanmar N = 18 Senegal N = 7 Uganda N = 10 Global N = 2 |

country coordinating mechanisms and the Global Fund secretariat and related decision-making bodies. To explore the drivers of decisions around RSSH investment systematically across countries, we drafted a list of likely root causes of high levels of investment in health systems support or strengthening and used these to structure the interview guide (S2 Text). Interview guides were then adapted by each country evaluation team to the specific country context. Interviewers took detailed notes during interviews which were transcribed afterward in preparation for analysis.

In both phases, respondents were selected purposively to include those involved in designing and/or implementing RSSH interventions and included health systems experts (at the global and country level), representatives of Global Fund principal- and sub-recipients, national program managers and other ministry of health leaders and implementing partners.

## Data analysis

Analysis of RSSH investments was undertaken by a working group comprised of a subset of evaluators from global and country evaluation partners. CEPs provided deep contextual knowledge of the Global Fund grants (including RSSH) in their respective countries, while GEPs added a layer of external validity, by working across multiple countries to ensure a consistent approach.

Coders reviewed each budget line item, including the module, intervention, activity description and Global Fund cost input categorization. Where more detail was required on activities included in budget files, we consulted funding request narratives (which accompanied the budgets as part of the submission materials) to provide additional description of activities. For each activity, we applied the three criteria (outlined in Table 1 above)—scope, longevity, and approach—to ascertain whether that activity could be considered supportive or strengthening to the health system. Budgets were coded independently by teams of three coders from each country, including GEP and CEP members. Coding teams then met to discuss any discrepancies and agree upon the final designation for each country. Final designation for an activity was determined based on whether a majority of the three criteria were support or strengthening (i.e., at least two out of three). However, in a limited number of cases, one or more of the criteria was unclear. Coding teams discussed these activities to arrive at consensus about whether to apply a final designation of supportive or strengthening. Determining whether to designate training activities as supportive or strengthening represents an example of one such area that required discussion between coders, as well as further review of application narratives to glean details on the content of the trainings. Lessons learned through coding were shared back iteratively with the larger working group during regular meetings to ensure external reliability across countries in how each team approached coding certain types of activities. After determining final designations for each activity, we quantified the proportion of funds allocated to health systems support and health systems strengthening in each budget, and further stratified the data by Global Fund budget module and intervention category to identify trends.

Notes from interview data were transcribed and drawing from the framework method [21], we developed an analysis matrix in Excel to support cross-country comparison and analysis. The matrix was organized by key question/theme (rows) and country (columns) and incorporated evidence from KIIs, document review, and budgetary analysis to support triangulation across data sources and across countries. Data was compiled from country specific analysis matrices that followed a similar structure of key question/theme (row) and respondent/stakeholder type (columns).

## Ethics statement

This analysis was conducted by two consortia, one led by IHME/PATH, another by Euro Health Group (EHG), and embedded within the work of the Global Fund Prospective Country Evaluations. For the IHME/PATH consortium, the University of Washington's Institutional Review Board (IRB) determined the PCE was non-human subjects research (#STUDY00003643), and research ethics review and approval was subsequently obtained from IRBs the DRC, Guatemala, Senegal, and Uganda, including the University of Kinshasa School of Public Health Ethics Committee in DRC (#ESP/CE/074/2017); the National Committee of Ethics in Health in Guatemala (#25–2017); the Université Cheikh Anta Diop de Dakar Research Ethics Committee in Senegal (#0345/2018/CER/UCAD); and the Makerere School of Medicine Research Ethics Committee (#REC REF 2017–146) and the Uganda National Council for Science and Technology (#SS 4472). Informed verbal consent was obtained from all interview participants and documented by interviewers on informed verbal consent forms, as approved by each IRB. For the EHG consortium, Ministry of Health approval was granted to conduct the evaluation in Cambodia, Mozambique, and Myanmar.

## Results

### Cross-cutting investment analysis

From the 2017–2019 (NFM2) to 2020–2022 (NFM3) funding cycles, we found that most PCE countries increased their overall RSSH budgets (see Fig 1). However, in both NFM2 and NFM3, a high proportion of funding went to activities that support rather than strengthen the health system and between funding cycles, the proportion invested in strengthening activities declined in three of the countries (DRC, Mozambique, and Uganda). In countries approaching transition away from Global Fund support, based on a combination of gross national income calculations and disease burden (e.g., Guatemala and Cambodia), our analysis indicates only marginal increases in the proportion allocated to strengthening the health system.

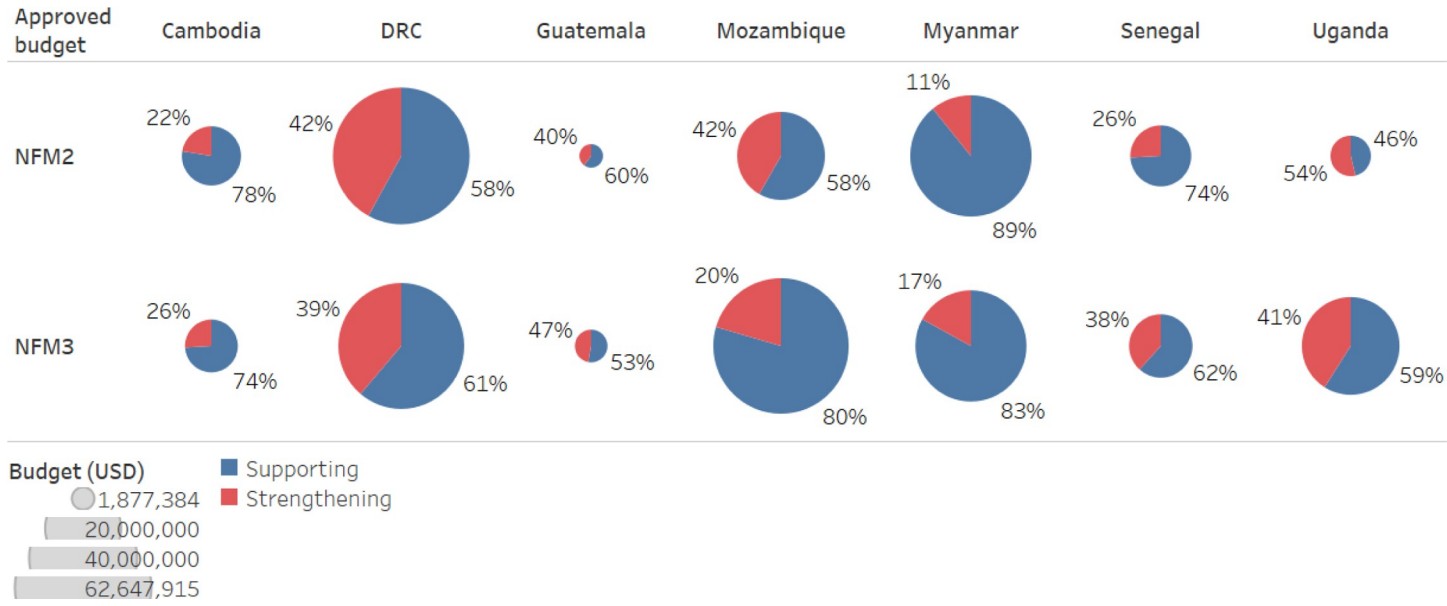

**Fig 1. Comparison of support or strengthening RSSH investments in NFM2 vs NFM3, by country.**

Interview respondents identified multiple reasons explaining these patterns and decisions to prioritize activities that are supportive in nature, rather than strengthening. Factors fell broadly into two categories: those associated directly with Global Fund policies and processes; and broader health systems and country contextual factors (Table 4).

Respondents noted that some activities designed to strengthen health systems (e.g., building data systems to facilitate evidence informed decision-making) require a longer time horizon to achieve results, often surpassing the three-year Global Fund grant cycle. Yet the Global Fund business model incentivizes high levels of budget utilization and implementation progress. In Cambodia, the prioritization of different types of RSSH activities for NFM3 was linked to performance in previous funding rounds, even though sub-optimal performance was frequently attributed to delays in disbursement of funds between principal and sub-recipients, not to successes or failures in implementing the activities. This contributed to high levels of investment in systems support, with the perception that demonstrating results from supportive investments would be easier within the three-year window of the grant. In multiple countries, while respondents confirmed the longer-term goal of moving toward strengthening investments, they also emphasized the need for supportive investments in critical inputs too, such as equipment, to build a foundation for future strengthening. For instance, a respondent in DRC emphasized the deeply intertwined nature of RSSH investments whereby a vehicle purchase, which is considered supportive, is a necessary input to enable supportive supervision. According to one respondent in Senegal, 'it's crucial to balance investments considering both aspects (support and strengthening) instead of arbitrarily pre-defining a dollar allocation for each of the two. For example, there is no point in acquiring equipment if the human resources are not trained.'

An added value of the Global Fund's approach to RSSH relative to other donors in the health systems space, according to many respondents, is the relative flexibility in allowing for RSSH funds to be used to address critical health systems gaps. And although most respondents understood conceptually the distinction between support and strengthening, they reported that operationalization is more challenging. Written guidance from the Global Fund during NFM3 did not contain explicit instructions on how to design interventions that contribute to strengthening rather than supporting systems. As such, respondents reported that the distinction did not factor into investment design decisions because there is no explicit requirement from the Global Fund to do so, or guidance on how to better design strengthening investments. Review of TRP reports of Global Fund NFM3 funding requests in PCE countries found very few comments related to RSSH, and none of these called for greater investment in strengthening activities. According to one respondent, not all TRP members consider themselves to be HSS experts, and they face pressure to keep to a minimum the number of issues to

**Table 4. Global Fund and contextual factors underlying decisions about RSSH investment.**

| | |
|---|---|
| **Global Fund business model factors** | • Three-year Global Fund grant cycle not well suited to longer-term, strengthening investments<br>• Minimal participation of health systems experts in regular Country Coordinating Mechanism meetings and in the funding request design process<br>• Lack of explicit guidance from the Global Fund on how to invest more strategically to strengthen health systems<br>• Minimal feedback during funding request review processes related to RSSH and the need to invest more in activities that contribute to strengthening in the longer term |
| **Contextual Factors** | • Weak sector-wide strategic planning for HSS, contributing to fragmented efforts by various disease programs and funders<br>• Inconsistent prioritization of HSS within MOHs, often driven by changes in leadership |

which they require principal recipients to respond as part of the application process. As such, RSSH is sometimes deprioritized during the funding request review process in relation to what are regarded as the primary aims of the Global Fund, namely ending HIV, tuberculosis, and malaria.

## Country-specific applied examples

A more detailed examination of RSSH investments in Guatemala, Senegal and Uganda helps to explain these patterns. In these countries, driven by interest of local stakeholders and the Global Fund Secretariat, the PCE conducted a more focused analysis of RSSH modules with a significant amount of Global Fund investment, and/or those where there was evidence of significant change from NFM2 to NFM3. The health management information systems/M&E module (HMIS/M&E) comprised the largest share of RSSH investment (in PCE countries, and for the Global Fund generally) whereas community systems strengthening (CSS) represented an area of growing investment and emphasis for the Global Fund.

**Guatemala.**   Fragmented and outdated information systems have long constrained the ability of the Guatemalan MOH, and other stakeholders, to report on program indicators and use data to inform decision making. In recognition of these challenges, since 2004 the Global Fund has provided funding to strengthen data systems, to support grant reporting in the near term and to contribute to integrated national information systems in the longer term.

Despite successive and sizeable investment in health management information systems and monitoring and evaluation (HMIS/M&E) across Global Fund grants, most of the funded activities are considered supportive. From NFM2 to NFM3 strengthening activities increased slightly from 46 to 47% (Fig 2). During NFM2, the majority of RSSH funding went to the HMIS/M&E module in the HIV grant, where the largest share was devoted to deploying DHIS2 to provide INCAP (the principal recipient for the HIV grant) with a system that allowed for timely reporting. At the time, the rationale was that this investment would also lay the groundwork for DHIS2 to be rolled out as the routine information system for the HIV program. However, despite initial buy-in, the MoH later expressed a preference for a licensed system, with consequences for the sustainability of the Global Fund investment in DHIS2.

Respondents reported that the high proportion of investment directed at supportive activities is attributable to a lack of sustained political will and wavering leadership around information systems strengthening, which has resulted in repeated investment by the Global Fund in different information systems over successive funding cycles, each with its own start-up costs.

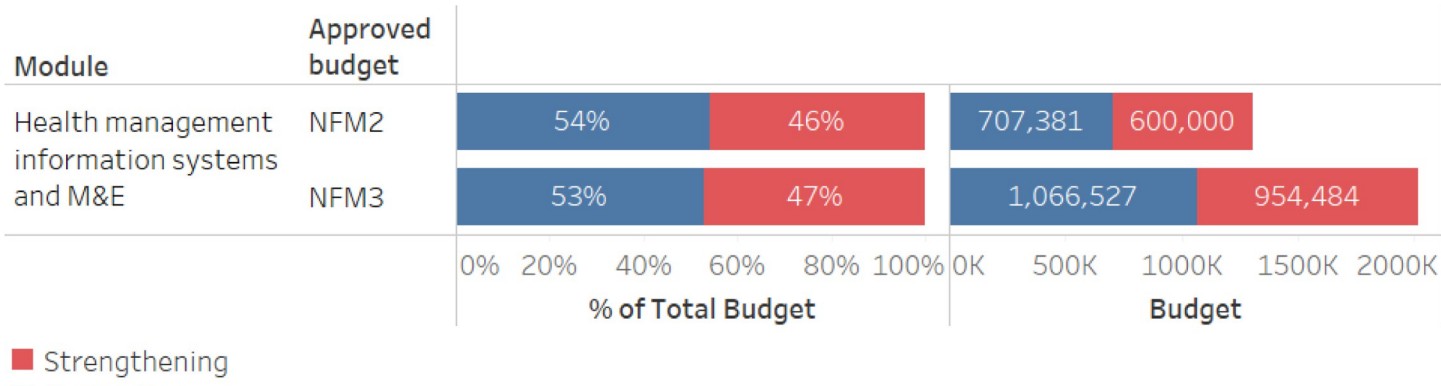

**Fig 2. HMIS investments in Guatemala, comparing NFM2 to NFM3, for total budget and proportion support vs. Strengthening.**

**Senegal.**   Senegal began roll-out of DHIS2 in 2014 with support from the Global Fund for various aspects of national scale-up. However, despite some improvement in the completeness and timeliness of data in DHIS2, concerns about data quality have constrained the use of DHIS2 data to inform decision making and program improvements. As such, as of 2020 all but the malaria program continued to employ parallel data systems, which could compromise the sustainability of the Global Fund's investment in DHIS2 in the longer term.

In NFM2, the HMIS/M&E investment was designed to be aligned to the objectives of the Digital Health Strategic Plan. However, much of the funding was directed toward fragmented, disease specific information systems, with just 6% devoted directly to DHIS2 and strengthening cross-cutting national systems. Supporting investments included equipping health facilities with tablets for data collection, training staff, etc. Additional funding was provided for disease-specific patient-level tracking systems in DHIS2, though the investment did not account for necessary activities such as data harmonization and data review meetings, community-level data collection, and disease specific program needs for reporting. As such, based on the 2S framework, the majority of NFM2 investment in HMIS/M&E was designed to support the health system. In NFM3, funding for strengthening activities increased within the HMIS/M&E module, both in proportional and absolute terms (Fig 3). This increase was driven by an expansion of cross-cutting activities to support the national HMIS, with ownership centralized under a single government principal recipient, rather than with each disease program. Rather than structuring data validation activities around data from fragmented disease specific systems, in NFM3 those processes would draw on data that adheres to standards in DHIS2 across programs, thereby helping to further institutionalize that system and a culture of data use in Senegal.

Despite this positive trend, there are concerns that continued reliance by disease programs on parallel data systems could undermine the long-term sustainability of DHIS2.

**Uganda.**   In NFM2, RSSH comprised a small proportion of the overall Global Fund investment in Uganda: 1.1% of the total budget, or US$5.5 million out of US$463.1 million. Of this, community systems strengthening (CSS) comprised 15% of the overall RSSH funds. Strengthening community systems has been identified by the Global Fund as essential to improving outcomes and the lack of robust community systems was found to be a crucial barrier especially to the achievement of tuberculosis treatment success.[22]

From NFM2 to NFM3, Uganda's investment in CSS increased over 8-fold from $830,071 to $6,889,800 (Fig 4). This increase was attributed to several factors. First, the Global Fund TRP

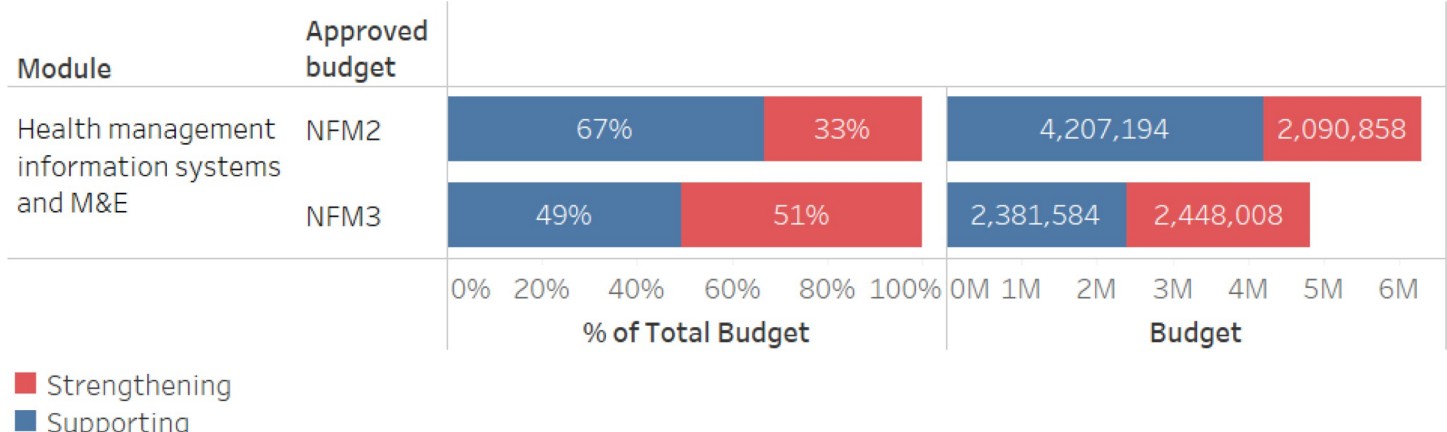

**Fig 3. HMIS investments in Senegal, comparing NFM2 to NFM3, for total budget and proportion support vs. Strengthening.**

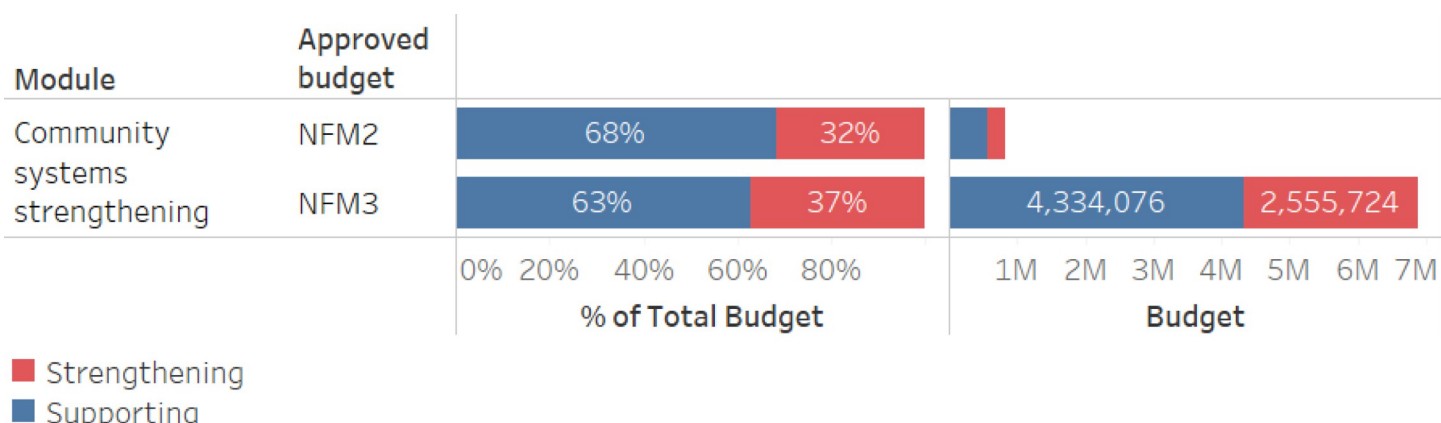

**Fig 4. CSS investment in Uganda, comparing NFM2 to NFM3, for total budget and proportion support vs. Strengthening.**

observations that highlighted the need for expanded community engagement through comprehensive and scaled activities in responding to the three diseases [23]. Second, evidence from the national CSS framework indicated that sub-national community-based organizations and networks had not benefited from previous CSS investments to build advocacy and social mobilization capacity. Third, lessons learned during NFM2 grant implementation, including recognition of the role of community scorecards in enhancing social accountability and promoting service delivery. As such, the substantial increase in the overall allocation to CSS funding during NFM3 corresponds to a rapid scaling of interventions in institutional capacity building, social mobilization, community-based monitoring, and community-led advocacy. Our analysis indicates the balance of supportive and strengthening investments in CSS remained similar between funding cycles with a slight increase in the proportion of funds allocated to strengthening, though in absolute terms the increase is much larger. Examples of activities categorized as strengthening include (1) stronger emphasis on CSO capacity strengthening; (2) inclusion of mentorship and training of district level advocates to conduct social mobilization activities; and (3) inclusion of orientation to/implementation of community scorecards coupled with deployment of digital technology for community-based monitoring of services.

To support more efficient RSSH implementation, in NFM3 Uganda proposed a new cross-cutting RSSH coordination structure detailing the role of principal recipients, and for CSS specifically, all cross cutting interventions were to be managed, coordinated, and implemented under the existing structures for the civil society principal recipient [24]. However, questions remain about the ability to implement a much larger CSS investment envelope effectively during NFM3, given that in NFM2, budget analysis indicated less than 30% absorption of CSS investments through two-and-a-half years into a three-year implementation period [24].

## Discussion

Our analysis found that from NFM2 to NFM3, there remained a high proportion of funds invested in activities that support, rather than strengthen health systems. This occurred despite calls by the Global Fund to invest more strategically to strengthen systems for health [16, 17]. Although during the design of NFM3 grants the 2S framework was not formally applied by those making decisions about how to invest in RSSH, or by the Global Fund TRP in deciding whether to approve those requests, our analysis indicates that the heavy and ongoing reliance on systems support means it is unlikely the Global Fund will achieve its stated objective of

building RSSH. We identified several factors underlying the high level of investment in support. First, health systems strengthening initiatives often require a long-time horizon and are therefore not well suited to the three-year grant cycle, especially given the emphasis on high absorption of funds and implementation progress. Furthermore, the level of investment by the Global Fund in RSSH is likely too small to sufficiently fund meaningful cross-cutting health systems strengthening in multiple areas. Second, the Global Fund is perceived as more flexible than other HSS donors in their willingness to fund inputs and fill critical systems gaps. Third, health systems experts were seldom deeply engaged in the RSSH design process. Fourth and finally, the Global Fund RSSH guidance, and the feedback received from the TRP in PCE countries, did not contain instructions around whether and how to invest more in strengthening.

Despite its stated intent to contribute to greater strengthening of health systems, the Global Fund allows for a great deal of flexibility to fill critical systems gaps. This represents an inconsistency in the business model as it fails to incentivize a shift toward strengthening, particularly in the face of persistent health systems funding shortfalls and resource constraints in many countries. This relative flexibility is however highly valued by those designing Global Fund investments which highlights a need to evolve the nature of investment from support to strengthening over time, especially as countries grow closer to transition from donor support alongside the maturation of their health systems.

Efforts to move toward meaningful contributions to strengthening health systems are also constrained by a lack of cohesion among donors sometimes including insufficient alignment with national government priorities. The Global Fund is just one of many actors in this space, albeit a large one, but health systems strengthening investments remain fragmented, often housed within the disease programs of the GHI providing the funding. While other GHIs such as Gavi require a bottleneck assessment to inform the design of its HSS investments, the cross-cutting nature of health systems strengthening suggests that a more comprehensive, system-wide landscape analysis and strategic plan for health systems strengthening (not specific to any single donor) would be beneficial [25]. This could inform the design of complementary investments by different donors all in service of broader systems strengthening objectives, moving beyond supporting just the immediate needs of disease programs.

The analysis presented in this paper examined whether Global Fund grants, as presented in final approved budgets by principal recipients and approved by the Global Fund, are *designed* to support, or strengthen the health system. Grants may or may not be implemented as designed. The PCE (among others) have identified multiple barriers to implementing Global Fund RSSH investments, all of which likely serve to constrain the impact of investments further. There is anecdotal evidence to suggest that large-scale, long-term initiatives to strengthen health systems may be more challenging to implement than shorter term investments (e.g., training or procurement of commodities) because of complexities inherent to implementation, including multi-sectoral coordination and cumbersome government procurement processes. Furthermore, outcomes and impact flowing from health systems strengthening efforts are notoriously difficult to quantify, which underscores a need for the development of a suite of qualitative and quantitative methods for measuring health systems strengthening [26]. However, there have been important recent advancements, for example through the work of the HSS Evaluation Collaborative, Bertone et al. (2022) identified 22 health system process goals representing features of a strong health system which can be operationalized for monitoring HSS [27]. Similarly, recent WHO guidance on measuring health systems resilience will help in defining clearer pathways between RSSH activities and associated outcome indicators, and help make the case for increased RSSH investment [28].

To this day, much debate persists internally at the Global Fund about their role in health systems strengthening, between a narrow approach in service of the elimination goals for the three diseases, versus a broad approach to contributing to strengthening primary health care systems in pursuit of universal health coverage. Similar tensions have played out at other GHIs, such as Gavi, surrounding the scope of health systems strengthening endeavors, with concerns raised that too much focus on HSS broadly could dilute their core, disease-oriented missions. All this notwithstanding, it is widely recognized that strong health systems are essential to achieving elimination goals for the three diseases. If the Global Fund is to play a role in building resilient and sustainable systems for health, improved guidance, and technical assistance on how to better design and implement HSS initiatives, paired with improved measurement of RSSH outputs and outcomes, are critical.

## Limitations

There are several limitations that should be considered when interpreting our results. First, this analysis was conducted by a team of evaluators representing global evaluation partners and each of the PCE countries. While having a different set of coders for each country provided deep contextual knowledge surrounding the content of Global Fund RSSH investments, it also presented challenges for inter-coder reliability across countries. As such, each country's RSSH budgets were coded by 1–2 members of the country evaluation partner, and one member of the global evaluation partner. Global evaluation partners worked across several countries to ensure consistency and the full working group met regularly to discuss how to code certain activities. Second, this analysis is focused on the design of RSSH investments, not on implementation or impact. The PCE was limited in its ability to measure the contribution of the Global Fund's RSSH investment to achieving its strategic objectives due to misalignment between the timing of the evaluation and that of funding cycles, and also by the paucity of intermediate indicators and performance framework indicators reporting on health systems outcomes [29]. Because the PCE ended in June 2021 shortly after approval of the NFM3 grants, we were unable to track implementation of these activities.

## Conclusion

Health systems strengthening remains an important priority in the Global Fund's 2023–2028 strategy [30]. M*aximizing people-centered, integrated systems for health to deliver impact, resilience and sustainability* appears as one of four mutually reinforcing contributory objectives, all serving the primary goal of ending HIV, tuberculosis, and malaria. Indeed, the COVID-19 pandemic has laid bare the need for more careful investment to strengthen emergency preparedness and response systems which requires a heightened emphasis on HSS. However, our analysis, along with a recent assessment by the TRP shows that these investments may not be designed to actually achieve such ambitious objectives [31].

Although this framework was not formally operationalized by the Global Fund Secretariat or grant recipients during NFM3, it remains critical to examine what Global Fund investments (and indeed, those of other donors) in health systems strengthening are actually designed to achieve [32]. More recent RSSH guidance from the Global Fund calls for applicants to place greater emphasis on systems strengthening and provide clear justification for investment in supportive activities. While this represents a positive development, there remains a need for detailed instruction and technical assistance on *how* to design interventions that contribute to long-term strengthening, but also further comparative analysis to determine whether this shift is merely rhetorical or contributes to substantive, measurable changes in the balance of supportive and strengthening investment.

The Global Fund identifies itself as the largest multilateral provider of funding for health systems strengthening [9, 11]. As emphasized by Chee et al. (2013), using health systems strengthening as a catch all term to describe any activities or interventions intended to address systems gaps or weaknesses, may serve to undermine such efforts in the long run as they are unlikely to achieve resilient and sustainable systems for health. This is likely to contribute to growing disillusionment by donors and others in seeking to mobilize much-needed resources to address such persistent systems issues.

## Supporting information

**S1 Text. PCE 2020 protocol on operationalizing the 2S framework.**
(DOCX)

**S2 Text. Key informant interview guide.**
(DOCX)

**S1 Checklist. Inclusivity checklist.**
(DOCX)

## Acknowledgments

This work would not have been possible without the many contributions of Global Fund stakeholders over the course of the evaluation. This includes ministries of health, non-governmental and civil society organizations in PCE countries, Global Fund Secretariat staff and other partners at the country and global levels. We are especially grateful to Sjoerd Postma, a former member of the TRP who introduced this framework at the Global Fund and advised the PCE team as we considered adapting it further. Thanks also to the TERG for their feedback and sustained support over the course of the evaluation period. Lastly, we acknowledge the full country and global evaluation teams, not all of whom are named as co-authors on this paper, but whose many contributions were instrumental.

**Disclaimer:** The authors' views do not necessarily reflect those of the TERG or the Global Fund.

## Author Contributions

**Conceptualization:** Nicole Salisbury, Saira Nawaz, Matthew Cooper, Katharine D. Shelley.

**Data curation:** Audrey Batzel, Emily Grapa, Francisco Rios Casas.

**Formal analysis:** Nicole Salisbury, Saira Nawaz, Justine Abenaitwe, Virginia Cerezo, Matthew Cooper, Ibrahima Gaye, Constant Kingongo, Louisiana Lush, Kate Macintyre, Eugene Manika, Shakilah N. Nagasha, Tidiane Ndoye, Rosario Orozco, Allison Osterman, Katharine D. Shelley.

**Funding acquisition:** Nicole Salisbury, Emily Grapa, Herbert C. Duber, Bernardo Hernandez, Katharine D. Shelley.

**Investigation:** Nicole Salisbury, Justine Abenaitwe, Virginia Cerezo, Matthew Cooper, Ibrahima Gaye, Constant Kingongo, Louisiana Lush, Kate Macintyre, Eugene Manika, Shakilah N. Nagasha, Tidiane Ndoye, Rosario Orozco, Allison Osterman, Katharine D. Shelley.

**Methodology:** Nicole Salisbury, Saira Nawaz, Justine Abenaitwe, Virginia Cerezo, Herbert C. Duber, Ibrahima Gaye, Bernardo Hernandez, Louisiana Lush, Kate Macintyre,

Eugene Manika, Shakilah N. Nagasha, Tidiane Ndoye, Rosario Orozco, Allison Osterman, Katharine D. Shelley.

**Validation:** Saira Nawaz, Justine Abenaitwe, Virginia Cerezo, Matthew Cooper, Herbert C. Duber, Ibrahima Gaye, Bernardo Hernandez, Constant Kingongo, Louisiana Lush, Kate Macintyre, Eugene Manika, Shakilah N. Nagasha, Tidiane Ndoye, Rosario Orozco, Allison Osterman, Katharine D. Shelley.

**Visualization:** Audrey Batzel, Emily Grapa, Francisco Rios Casas.

**Writing – original draft:** Nicole Salisbury.

**Writing – review & editing:** Saira Nawaz, Justine Abenaitwe, Audrey Batzel, Emily Grapa, Francisco Rios Casas, Virginia Cerezo, Matthew Cooper, Herbert C. Duber, Ibrahima Gaye, Bernardo Hernandez, Constant Kingongo, Louisiana Lush, Kate Macintyre, Eugene Manika, Shakilah N. Nagasha, Tidiane Ndoye, Rosario Orozco, Allison Osterman, Katharine D. Shelley.

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
