## [Decision Letter · Decision Letter 0]

19 Mar 2024

PGPH-D-23-02388

Ambitions and realities: are Global Fund investments designed to achieve resilient and sustainable systems for health? Findings from the Global Fund Prospective Country Evaluation

Dear Dr. Salisbury,

Thank you for submitting your manuscript to PLOS Global Public Health. After careful consideration, we feel that it has merit but does not fully meet PLOS Global Public Health’s publication criteria as it currently stands. Therefore, we invite you to submit a revised version of the manuscript that addresses the points raised during the review process.

We look forward to receiving your revised manuscript.

Kind regards,

Karen A. Grépin, Ph.D.

Academic Editor

Journal Requirements:

1. Please include a complete copy of PLOS’ questionnaire on inclusivity in global research in your revised manuscript. Our policy for research in this area aims to improve transparency in the reporting of research performed outside of researchers’ own country or community. The policy applies to researchers who have travelled to a different country to conduct research, research with Indigenous populations or their lands, and research on cultural artefacts. The questionnaire can also be requested at the journal’s discretion for any other submissions, even if these conditions are not met.  Please find more information on the policy and a link to download a blank copy of the questionnaire here: https://journals.plos.org/globalpublichealth/s/best-practices-in-research-reporting. Please upload a completed version of your questionnaire as Supporting Information when you resubmit your manuscript.”

2. In the ethics statement in the Methods, you have specified that verbal consent was obtained. Please provide additional details regarding how this consent was documented and witnessed, and state whether this was approved by the IRB.

Additional Editor Comments (if provided):

Dear Authors,

After considering the reviews of both reviewers and the especially helpful feedback from reviewer 2. I would invite you to consider the feedback below on how to strengthen the manuscript, especially with regards to the justification of the framework used to characterise the investments.

Reviewers' comments:

Reviewer's Responses to Questions

**Comments to the Author**

1. Does this manuscript meet PLOS Global Public Health’s publication criteria? Is the manuscript technically sound, and do the data support the conclusions? The manuscript must describe methodologically and ethically rigorous research with conclusions that are appropriately drawn based on the data presented.

Reviewer #1: Yes

Reviewer #2: Yes

2. Has the statistical analysis been performed appropriately and rigorously?

Reviewer #1: N/A

Reviewer #2: I don't know

3. Have the authors made all data underlying the findings in their manuscript fully available (please refer to the Data Availability Statement at the start of the manuscript PDF file)?

Reviewer #1: Yes

Reviewer #2: Yes

4. Is the manuscript presented in an intelligible fashion and written in standard English?

Reviewer #1: Yes

Reviewer #2: Yes

5. Review Comments to the Author

Reviewer #1: This manuscript is intelligently presented and written in standard English.It is technically sound and provides data that supports the conclusion.In addition, the manuscript was methodologically and ethically appropriate and met the publishing criteria of PLOS Global PublicHealth.

Reviewer #2: Thank you for giving me the opportunity to review this manuscript. I found the paper to be both an exciting and valuable analysis. The analysis has the potential to make a significant contribution to the discourse on the role of global health initiatives in strengthening health systems, especially in low- and middle-income countries. While I have a few comments, they mainly pertain to the need for additional information to supplement the arguments and analysis and better presentation of data for clarity. However, I have three main concerns with this analysis, concerning conceptual clarity and methodological rigour, especially from generalising these findings for other global initiatives.

I have three major concerns and a few minor comments.

1. My first concern is the use of a more than a decade-old support/strength framework in the analysis. While this framework is important, it emerged at a time when there wasn't much clarity about what HSS is and when HSS was gaining popularity as the new buzzword among global health actors. The discourse of HSS has expanded significantly in the past ten years to call out the dichotomies between support and strengthen as false dichotomies, issues around such objective black and white distinction between interventions, and stressing the importance of the support interventions.

Ignoring the evolution of the concept of HSS in the last decade, a fleeting mention that such black or white objective framing of HSS is not the best framing shows poor engagement with the idea of HSS. There is scope to give a thorough justification for using the framework, especially when the use of this framework is complemented by other methods of collecting data and perspectives of stakeholders through interviews.

For example, around Lines 65-66, “While both are necessary, Chee and colleagues emphasize that failing to invest in strengthening activities may undermine calls for more investment in HSS, because if designed to support and not strengthen, HSS investments are unlikely to achieve their objectives.”, there is scope for giving some context about why support interventions are not perceived equal to strengthening while they fulfil more urgent/acute gaps. Lines 242- 247 highlight this issue with an apt example of purchasing vehicles, and I was expecting a deeper engagement with such conflicts and grey areas while choosing an analytical framework for an HSS evaluation.

2. The second and related concern is the insufficient engagement with key terms like HSS, resilient, sustainable systems, etc, at both the conceptual and practical levels. Readers of an HSS evaluation paper would expect some background on the meaning of these terms, the debates around their meaning, and the shifts in discourse (if any) from HSS to resilience/sustainable systems, as well as the context of this shift.

I was looking for the description/definition of RSSH or a few words about how Global Fund defines RSSH. Some background of why Global Fund started to invest in HSS and why HSS became an important investment for Global Fund considering it is a disease-specific initiative.

For example, Lines 72-74“The prominence of HSS in relation to the Global Fund’s organizational mission has increased over the last decade, and in its 2017-2022 strategy , building resilient and sustainable systems for health (RSSH) was one of four strategic objectives.” Clarifying this meaning/definition/framework is also important because Line 117 you mention Total RSSH funding is summarized across a country’s grants, and includes “direct” RSSH only, meaning investments that were categorized as RSSH per the Global Fund’s modular framework definitions.”

Similarly, line 406-407, “debate persists internally at the Global Fund about their role in health systems Strengthening” gives an opportunity to expand the argument of the (limited or wide) opportunities that disease specific global initiatives present for HSS.

3. Third is the presentation of methods section. There are several places to improve the methods section. Line 105 states that this analysis is embedded within the larger PCE platform and relied upon a subset of the methods described in Box 1. Few more details about these methods would be useful here.

Giving a matrix/table of different methods used in this analysis and how they helped meet the objectives would be useful. For example, a list/table of all quantitative and qualitative methods and the data sources for these methods in the table ( like for interviews stating the number of each category of respondents in each country) would be useful.

Perhaps, creating sub heads for different kinds of data collected and their significance will bring more clarity. Explaining the significance of two rounds of interviews would be useful.

Table 2- Adding an additional column of example interventions (even if hypothetical) would be bring more clarity in differentiating support and strengthen interventions.

Minor comments

• Line 181 line- Adding more details about how you maintained consistency across teams would be useful.

• Line 109: “The analysis of RSSH investments was conducted in seven PCE countries”- Did you mean eight countries?

• Line 386 “Efforts to move toward meaningful contributions to strengthening health systems are also constrained by a lack of cohesion among donors”. A few words about national/government priorities would be a good addition here ( it is also an important finding if your respondents did not bring the government and national actors in these discussions).

• Line 183-185 giving an example of such coding activity will be useful.

• Line 225- explaining in a few words what you mean by contextual factors would be helpful.

• Line 241- explaining what you mean by supportive investment would be helpful.

• Line 266- explaining why you did a deep dive in only 3 countries would be useful.

• Abstract- readers might not know Chee's argument on support Vs strength. Adding a sentence in differentiating the two would be helpful.

6. PLOS authors have the option to publish the peer review history of their article (what does this mean?). If published, this will include your full peer review and any attached files.

**Do you want your identity to be public for this peer review?** For information about this choice, including consent withdrawal, please see our Privacy Policy.

Reviewer #1: **Yes: **Dr Shazia Sajjad Sarhandi, MBBS, MPH, MPA.

Reviewer #2: **Yes: **Sumegha Asthana

---

## [Editor Report · Decision Letter 1]

18 Oct 2024

Ambitions and realities: are Global Fund investments designed to achieve resilient and sustainable systems for health? Findings from the Global Fund Prospective Country Evaluation

PGPH-D-23-02388R1

Dear Dr. Salisbury,

We are pleased to inform you that your manuscript 'Ambitions and realities: are Global Fund investments designed to achieve resilient and sustainable systems for health? Findings from the Global Fund Prospective Country Evaluation' has been provisionally accepted for publication in PLOS Global Public Health.

Best regards,

Karen A. Grépin, Ph.D.

Academic Editor

Dear Authors,

Thank you for incorporating the reviewer's excellent feedback into your revised manuscript. As most of the feedback had been with regards to being clearer with regards to terminology and more explicit with regards to methods, I believe that the changes that you have made have sufficiently addressed the feedback of the reviewer. I now believe that the article is ready for publication. I thank you for submitting your article to PLOS Global Public Health.